# Superstructure Detection in Nucleosome Distribution Shows Common Pattern within a Chromosome and within the Genome

**DOI:** 10.3390/life12040541

**Published:** 2022-04-06

**Authors:** Sujeet Kumar Mishra, Kunhe Li, Simon Brauburger, Arnab Bhattacherjee, Nestor Norio Oiwa, Dieter W. Heermann

**Affiliations:** 1Institute for Theoretical Physics, Heidelberg University, D-69120 Heidelberg, Germany; sujeetsankrityan@gmail.com (S.K.M.); li@thphys.uni-heidelberg.de (K.L.); brauburger@stud.uni-heidelberg.de (S.B.); oiwa@thphys.uni-heidelberg.de (N.N.O.); 2School of Computational and Integrative Sciences, Jawaharlal Nehru University, New Delhi 110067, India; arnab@jnu.ac.in; 3Department of Basic Science, Universidade Federal Fluminense, Nova Friburgo 28625-650, Brazil

**Keywords:** chromatin, nucleosome positioning, nucleosome distribution, heterochromatin, euchromatin, structure classification

## Abstract

Nucleosome positioning plays an important role in crucial biological processes such as replication, transcription, and gene regulation. It has been widely used to predict the genome’s function and chromatin organisation. So far, the studies of patterns in nucleosome positioning have been limited to transcription start sites, CTCFs binding sites, and some promoter and loci regions. The genome-wide organisational pattern remains unknown. We have developed a theoretical model to coarse-grain nucleosome positioning data in order to obtain patterns in their distribution. Using hierarchical clustering on the auto-correlation function of this coarse-grained nucleosome positioning data, a genome-wide clustering is obtained for *Candida albicans*. The clustering shows the existence beyond hetero- and eu-chromatin inside the chromosomes. These non-trivial clusterings correspond to different nucleosome distributions and gene densities governing differential gene expression patterns. Moreover, these distribution patterns inside the chromosome appeared to be conserved throughout the genome and within species. The pipeline of the coarse grain nucleosome positioning sequence to identify underlying genomic organisation used in our study is novel, and the classifications obtained are unique and consistent.

## 1. Introduction

The genomes of all higher eukaryotes are organised in different structures on multi-length scales [1,2]. Of these organisational structures, the chromosome is the biggest one, being observable under a normal light microscope. The smallest organisational structure, one level above the double helix DNA, is the nucleosome where 147 base pairs (bp) of DNA are wrapped 1.65 times around a histone octamer [3,4,5]. The arrays of nucleosomes organise to form the chromatin fibre, which folds into two mutually excluded structural domains, namely “heterochromatin” and “euchromatin”. The “heterochromatin” regions are enriched with inactive/repressive genes and are usually positioned closer to the periphery of the nucleus. The “euchromatin” regions contain transcriptionally active chromatin [3,6,7], which are genes located in the interior of the nucleus. The hierarchical packaging of chromatin renders the genome a very compact conformation that provides controlled accessibility of the regulatory DNA sequences (genes) by other DNA-binding proteins (DBPs) [8,9]. Thus, the chromatin organisation is tightly linked to gene regulation and warrants detailed investigation. Various experimental techniques have been developed to probe the hierarchical chromatin organisation at different length scales. For instance, the “chromatin conformation capture” experiment (e.g., 3C and HiC) [2,10,11] captures the organisation of chromatin in a kbp to Mbp length scale, revealing the formation of topologically associated domains (TADs) [12] and chromatin loops [13,14]. Further characterisation of the chromatin fibre at the length scale of genes (∼kbp) is achieved by the Micro-C technique that captures the intra-chromatin interactions at a resolution of ∼100 bp within an organisation module called chromosomal interaction domains (CIDs) [15,16]. CIDs are much smaller but still similar to TADs. These structural organisations are strongly regulated by the nucleosome positions, length of linker regions, and presence of nucleosome-depleted regions (NDR) across the chromosome [17].

The term “nucleosome positioning” refers to the location of nucleosomes along the sequence of genomic DNA. Nucleosome positioning is determined by several factors, including DNA sequence [18,19], DNA-binding proteins [20,21], nucleosome remodelers [22,23,24], RNA polymerases [25], and more. Although nucleosome positioning is a dynamic process, the sequence-based mapping approach identifies its position only in a cell- and time-averaged manner. The technology of micrococcal nuclease (MNase) digestion combined with high-throughput sequencing (MNase-seq) [26] is a powerful method to map the genome-wide distribution of nucleosome positioning and its occupancy. The resulting occupancy maps are ensemble averages of heterogeneous cell populations and may also be influenced by titration [27]. However, it is necessary to retrieve the cell-specific features from the population average to reveal the mechanism of nucleosome organisation and its translocation along the genome. Zhang et al. has developed an algorithm called “Nucleosome Positioning from Sequencing” (NPS) to predict accurate nucleosome positioning from the MNase-seq data, which was later improved to iNPS (improved NPS) [28]. The nucleosome positioning here is considered as an average static picture where they implicitly consider the nucleosome dynamics in the form of snapshots at different time- and cell-averages. This nucleosome positioning provides the frequency of its occurrence from which peaks are annotated to obtain possible nucleosome location along the sequence. In short, the nucleosome positioning data from iNPS are simply the most probable nucleosome position along the chromosome. Furthermore, extensive studies have been performed to recognise nucleosome positioning patterns around CTCFs, transcription start sites (TSSs), exons and introns, promoter and loci regions locally. For instance, a typical nucleosome distribution around TSSs indicates nucleosome depletion, resulting in a nucleosome-free region (NFR), whereas the nucleosomes downstream of TSS are equally spaced [29]. A similar observation around CTCF is obtained: an array of well-positioned nucleosomes flank the sites occupied by the insulator binding protein CTCF across the human genome [30]. Despite the efforts, the global picture of nucleosome positioning remains elusive until a recent study that has reported three types of nucleosomal arrangement by analyzing the nucleosome spacing and phasing in a genome [31]. The evenly spaced nucleosomes in the array are termed as a regular array and irregular otherwise. At a given genomic location in the cell population, nucleosomes may also assume similar positions and are referred to as phased arrays. The phased-regular nucleosome arrays, being most prominent, are the hallmark of chromatin and found to be conserved from yeast to mammals. These phased-regular nucleosome arrays are mostly found near the promoter regions of transcribed genes in the yeast genome and near the binding sites of high-affinity DBPs in higher eukaryotes. However, the findings have limited applicability only at local regions of the chromatin fibre and provide absolutely no information about the nucleosome organisation along a complete chromosome or genome.

We used a theoretical approach to obtain a novel classification of segments across the chromosome based on the similarity in nucleosome patterns. The nucleosome positioning data are used as inputs that are systematically coarse-grained to analyze their auto-correlation function to search for any pattern. The results are processed using hierarchical clustering techniques to investigate if there exists any unique pattern of nucleosome. Our results suggest that the positions and occupancy of nucleosomes in a chromosome are not random; rather, they reveal distinct patterns of distribution within a chromosome. Interestingly, the patterns appear to be conserved within the genome as well and are in agreement with the previous study that has reported three distinct nucleosome organisations across the genome. Furthermore, at the chromosome level, our approach could capture a few unique patterns in the range of the ∼50 kbp length scale, which repeatedly occur throughout the chromosomes, indicating they might play a crucial role in regulating gene networks at a more local scale. The study underpins the nucleosome positioning architecture inside a genome that can provide insights into the genome organisation (c.f. Figure 1) not known before.

### 1.1. Data

The technology of micrococcal nuclease (MNase) digestion combined with highthroughput sequencing (MNase-seq) [26] is used to map the distribution of nucleosome occupancy genome-wide. In order to map the MNase-seq data to nucleosome positioning data, several programs were developed, such as NPS [32], nucleR [33], and DANPOS [34]. A nucleosome sequencing profile is generated to depict nucleosome distribution in wave-form where nucleosome peaks are detected. The improved nucleosome-positioning algorithm (iNPS) can be applied to identify peaks and correctly detect nucleosome positions [28]. One possible output of the iNPS algorithm is in the binary format, with 1s representing a nucleosome being present and 0s for the nucleosome-free regions or linker regions.

The genome-wide study of the species is a challenging task due to its large sequence size, which needs theoretical expertise and computational power. For our study, we have chosen *Candida albicans* as a simple completely sequenced organism [35] that is small enough to be computationally viable. Furthermore, *C. albicans* allows for similar mechanisms that are found in eucaryotes. Indeed, epigenetic mechanisms across animals, plants, and fungi include DNA methylation as a common epigenetic signalling mechanism, and it is present in *C. albicans*. A putative histone H1 has been identified [36]. Whereas these are technical decisions, we also wanted to select a species that should have a clinical prevalence. It consists of eight sets of chromosome pairs whose complete genome sequence is available. The raw data of the MNase-seq are available from the Gene Expression Omnibus (GSM1542419) and were measured by Puri et al. [37]. We also accessed the processed iNPS data in the NucMap database by Zhao et al. [38].

### 1.2. Methods

To obtain a consistent classification of the nucleosomal positioning data in genomewide classes, we perform the following steps (explained in more detail below the list):Each chromosome is divided into segments of 75 kbp of length.For every chromosome, the positioning data are coarse-grained.The coarse-grained nucleosome positioning data are used to calculate auto-correlation functions over the different sections.A distance matrix is calculated over all the auto-correlation function data.These segments are clustered. Various distance matrix and clustering algorithms are used to generalize the results.

#### 1.2.1. Genome Section Classification

In order to extract the global pattern for areas in a genome, the whole genome is separated into sections with equal length. The section length *L* is an important scale parameter and needs to be properly set. *L* should not be too large to avoid all features from different areas bounded together. At the same time, *L* also should not be too small; otherwise, the global structure is flooded by the subtle differences and becomes a pattern for only a single nucleosome. The single nucleosome wrapping length $L_{n}$ can be used as a lower bound for the choice of *L*. However, to obtain a relevant structure, we require that $L>>L_{n}$. Considering the nucleosome length $L_{n}$ is about 147 bp [3,4], *L* is chosen to be 50 kbp. Additionally, to avoid boundary effects, for each section, a 12.5 kbp intersection on both sides with its neighbor is added. Hence, the total section length *L* is 75 kbp. This binning is applied to each chromosome. Chr. 2 for example, with a length of 2,231,883 bp, is separated into 44 sections.

#### 1.2.2. Coarse Graining

The idea of coarse graining is an established ansatz and tool in physics to describe complex systems on a scale that allows identifying structure. Typically, the structure appears as a collective phenomenon among smaller entities. The idea is to eliminate degrees of freedom, i.e., find a representation of the system on a larger time or space scale, iteratively moving to larger scales without changing the system. Over the last few years, coarse graining has emerged as a way to model large complex systems and has successfully been applied to other biomolecules such as proteins [39].

After the whole genome is separated into sections, coarse graining is applied for each section. The method we implemented for coarse graining is the rolling mean method [40]. This method takes a window with a certain size (e.g., b=5 kbp), computes the averaged value of the nucleosome positioning inside the window, and moves the window to the following location. After this value is computed for each location, coarse-grained data on the scale of the window size are returned. Here, Python pandas.DataFrame.rolling [41] is used to obtain the coarse-graining. To exclude the effect of telomeres, discrete ends of the sections and incorporation of the window size and offset was chosen to be at least
(1)offset≥windowsize/2

#### 1.2.3. Auto-Correlation Function Calculation

An auto-correlation function is a well-known approach in physics and pattern recognition, capturing the inner interaction pattern inside the data [40]. Particularly for structures that are liquid-like, the auto-correlation function, or in this context the radial distribution function, identifies typical length scales and patterns.

For each section *j*, it is applied on all the coarse-grained data $\rho_{j}$. The normalized auto-correlation function $C^{j}\left(\tau \right)$ with respect to distance τ for section *j* is:(2)$$C^{\alpha ,j}\left(\tau \right)=\frac{E[\left(\rho_{i}^{\alpha ,j}-\mu^{\alpha ,j}\right)\left(\rho_{i+\tau }^{\alpha ,j}-\mu^{\alpha ,j}\right)]}{{\left(\sigma^{\alpha ,j}\right)}^{2}}$$
where $\rho_{i}^{\alpha ,j}$ is the data at position *i* within the section *j* of chromosome α. E(⋯) is the mean of everything in the parentheses over all indices *i*. $\mu^{j}$ is the mean of ρ and $\sigma^{j}$ is the variance for the section *j*. Thus, associated with each section *j* is the function $C^{\alpha ,j}\left(\tau \right)$ of chromosome α; hence, at the end, we will have *N* functions $C^{\alpha ,j}\left(\tau \right)$ where *N* is the section number for the particular chromosome.

#### 1.2.4. Distance Matrix Calculation

To classify the functions, a similarity measure is applied, and a resulting distance matrix is computed. The distance matrix is a square matrix containing the pairwise distances between all the elements available in the dataset, measuring the proximity between the correlation functions. Interpreting the functions as high-dimensional vectors, we use the *p*-norm to define the distance $d_{p}$ between two functions:(3)$$d_{p}\left(a,b\right)={∥a-b∥}_{p}={\left(\sum_{i=1}^{d}{\left|a_{i}-b_{i}\right|}^{p}\right)}^{1/p}$$
where *a* and *b* are the functions in the form of vectors. For p=2, the *p*-norm corresponds to the Euclidean distance.

#### 1.2.5. Clustering

To identify the unique nucleosome organisation or distribution function, there is a need to cluster the sections together on the basis of similarity among them. We used a clustering approach, i.e., hierarchical clustering [42]. This is an unsupervised algorithm that groups similar objects into groups called clusters. It uses a distance matrix to identify the two closest clusters first and then merge the two most similar clusters. This iterative process continues until the clusters are merged to get distinct clusters in a hierarchical manner.

Hierarchical clustering builds a hierarchy of clusters using two methods: agglomerative and divisive algorithms. We used the former, i.e., the Ward method [43], where each observation starts in its own cluster and pairs of clusters are merged, moving up the hierarchy.

#### 1.2.6. Statistical Distributions Fitting

Fitting of the distributions was performed using the scipy stats package [44] under Python.

## 2. Results

The first indication of non-trivial ordering is given by the distribution of the nucleosome positioning data. The binary nucleosome positioning data for all chromosomes of *Candida albicans* (NucMap database [38]) are subjected to the described coarse graining and then analyzed (see the histogram of densities in the Appendix A). The genome-wide normalised nucleosome density shows a non-Gaussian behaviour with a slight negative skew. Overall, a log-logistic distribution gives the best consistent fit for all chromosomes compared to a normal distribution on the same bin size and rolling average for all chromosomes.

Recall that each chromosome is divided into chunks of 75 kbp with 25 kbp overlapping on each side. The auto-correlation of each chunk is obtained on the coarse-grained nucleosome positioning data. The respective correlation function of each section for all chromosomes are shown in Figure 2 and in detail in Appendix A. Shown are the correlation functions on the coarse-grained scale as well as a further smoothing to make the features that are common among a class more apparent (see below). The colour bar indicates the class. Even though there are variations within a class, certain common features are seen. These features are the first and second peak structure, the height of the peaks, and how long a structure persists. Recall that the zero line indicates that there is no correlation; i.e., there, the structure is that of a gas or an unordered behaviour. The first peak indicates an increased probability to find a coarse-grained nucleosome at the distance of the peak position, and the same applies to the second and additional peaks. If these peaks are of similar height, then there is a stronger long-range ordering. A particular example showing similar heights up to a third peak is in section 12 of chromosome no. 3 (see Appendix A), while section 6 shows a drop in the peak heights. Nevertheless, due to the overall similarity, these fall into the same class.

With diminishing height, the likelihood of the ordering and the strictness of ordering vanishes. Notice that for some of the sections (within one class), many sub-peaks or side-peaks exist, indicating possible sub-orderings. An example on the more extreme side is chromosome 3 and sections such as 3,5,16, etc. Overall, the short-range order is much less pronounced. The orange smoothed line indicates that in this class, the salient feature is a smoothly decreasing function indicating a different kind of order than for the class with sections 0, 8 and 12, etc.

Even looking at the correlation functions without the indicated class mapping shows that there are universal features beyond fluctuations. Within a class, a more or less pronounced ordering feature is visible. Comparing the different correlation data between the chromosomes, these become apparent.

These observations can be proven more rigorously by applying similarity measures between the correlation functions. Figure 2 shows the resulting distance matrix between all chromosomes and all sections (the individual results are shown in the Appendix A). Shown is the distance matrix after reordering on the basis of similarity between sections. The colour indicates the similarity between the correlation functions. Notice the patterns that emerge from the sorting of the data into classes.

These classes, represented by different colours, are shown in the dendrogram. These classes were obtained by hierarchical clustering. In the lower part of the figure on the left are the typical correlation functions representing the corresponding class with its colour code. The orange-coloured class shows a fairly regular pattern and closely spaced ordering on a short scale, such as tightly packed heterochromatin, whereas the light blue class has lost the regularity and shows a less stringent regular but still pronounced pattern on a slightly larger scale. The blue-coloured class shows a rather very irregular pattern compared to the other two classes and corresponds more to euchromatin.

These observations are consistent with the typical classification from microscopy data into hetero- and euchromatin. The data show that the orange and light blue classes can be mapped on heterochromatin. Thus, the blue-coloured class is euchromatin. The data also show that still, within any of these classes, the features have many sub-features that we salvaged for the larger patterns to allow a “coarse-grained” view on the ordering of the nucleosomes. These sub-features compose elaborated chromatin states such as solenoid [45], zig-zag ribbon [46], or other structures [47], which demand a cross-correlation analysis with CTCF binding sites [48], CpG island position [49], and other data.

Notice that this partitioning into classes is genome-wide. A consistent classification can be established. This is shown in the mapping of the positions of the section to the chromosomes. Notice that, as expected, not a random mixture of the three colours emerges but rather a clear pattern. The larger chromosomes appear to have more internal structuring compared to the smaller chromosomes that are more homogeneous in their internal structure. The partitioning into a clear pattern, genome-wide is not limited to species *Candida albicans*, but the pipeline is generalised and can be used for any species in which the whole genome has been sequenced.

## 3. Discussion

The structural organisation of the genome depends on the patterns of nucleosome positioning and their distribution in the genome. At a higher scale, the nucleosome positioning distribution varies across the chromosomes, which appear to be conserved along the entire genome. The classification of the chromosomes into segments of the distinct nucleosomal distribution shown here is in line with earlier studies. Although two major classifications of the chromosomal region as heterochromatin and euchromatin are suggested, we find that their organisations can be further subdivided. Nucleosomes can be well-positioned to form phased and unphased arrays consisting of regularly spaced nucleosomes or can be fuzzy to form irregular arrays of nucleosomes. The three distinct nucleosome distribution patterns along the genome obtained in our result are in agreement with this study. Moreover, further classification of nucleosomal distribution is obtained along each chromosome. Around five to seven different nucleosome distribution patterns are observed for all chromosomes. However, for the entire genome, three patterns are found to be conserved.

We have analysed the effect for different p=2,7 in the *p*-norm on the outcome of the clustering of similar correlation functions, and the outcome comes to be similar for all *p*. For high *p* values, some of the clusters split into further clusters. In addition, the cosine similarity norm was tested for further verification, yielding similar clustering (see Appendix A). This rules out that the clustering is an artifact of the model and its architecture.

Around five patterns of chromosomal organisation are obtained for each chromosome by analysing the nucleosome positioning data distribution. These patterns obtained are generally coincident with gene densities and lead to the distinct spatial organisation of genomic DNA. The genome’s hierarchical structure–function relationship [12] is governed by chromatin domains and their higher-order folding. The formation of chromatin boundaries and associated TADs are controlled by the nucleosome distribution patterns. Recent studies by Wiese et al. [16] suggested that domain formation and genome organisation can be predicted with nucleosome positioning only. Pulivarty et al. [50] primarily focused on nucleosome studies, which are limited to a very local individual promoter and enhancer but can be a more general mechanism by which cells can regulate the accessibility of the genome during development at different scales. After an extensive analysis of nucleosome positioning data, the way of organisation of nucleosomal distribution patterns is found to be different at different scales and for different chromosomes. The distinct patterns obtained from our calculation correspond to different ways of nucleosome positioning and may control domain formation and genome organisation in the cell. However, the three distinct patterns of nucleosome organisation that appeared to be conserved in the genome show the global consistency of distribution patterns inside the genome. The consistency in different kinds of distinct patterns observed in the genome corresponds to identical gene densities and similar expression regions for specific locations inside the cell.

## Figures and Tables

**Figure 1 life-12-00541-f001:**
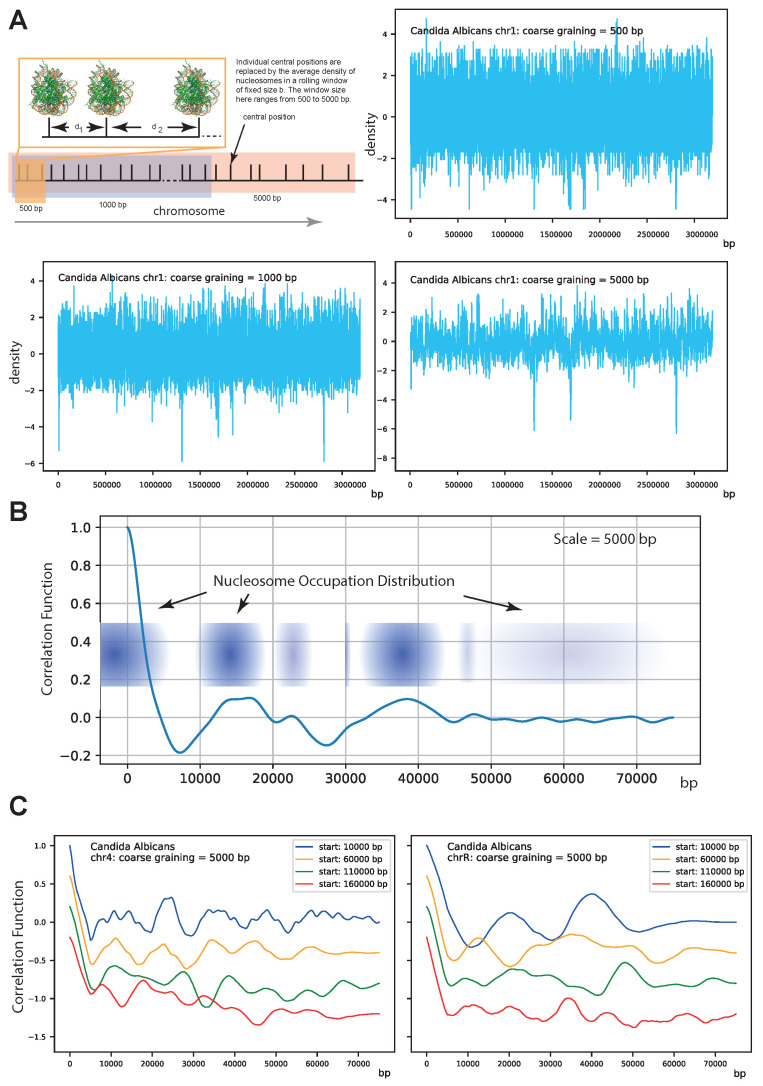
(**A**) shows the performed coarse-graining procedure and results for coarse-graining lengths *L* of 500 bp, 1000 bp, and 5000 bp. More structure is visible as *b* is increased. Going up even further washes out the structure. This is typical for systems with an intrinsic length scale. (**B**) shows the correlation among the coarse-grained super nucleosomes. The structure is that of a system exhibiting short range-order that is liquid-like with first and second nearest neighbor peaks. If there is no order or correlation, then the correlation function would be constant. On the other hand, if one would see strong regular peaks, this would indicate a regular ordering with the peak distances giving the preferred distance between the coarse-grained nucleosomes. The oscillatory characteristic with a larger first peak and smaller second peak indicates that two coarse-grained nucleosomes are on average located within a distance from the origin to the first peak and a second coarse-grained nucleosome at the distance indicated by the second peak. Since the peaks are decreasing, this ordering diminishes, much like the local ordering in a liquid. On larger scales larger than 50,000 bp, there is no order, i.e., there is no correlation. (**C**) shows for two chromosomes how the structure differs within as well as among chromosomes. The parameter start indicates from where in the chromosomes the structure was computed. One can see that the structure varies within a chromosome; nevertheless, common structures are found.

**Figure 2 life-12-00541-f002:**
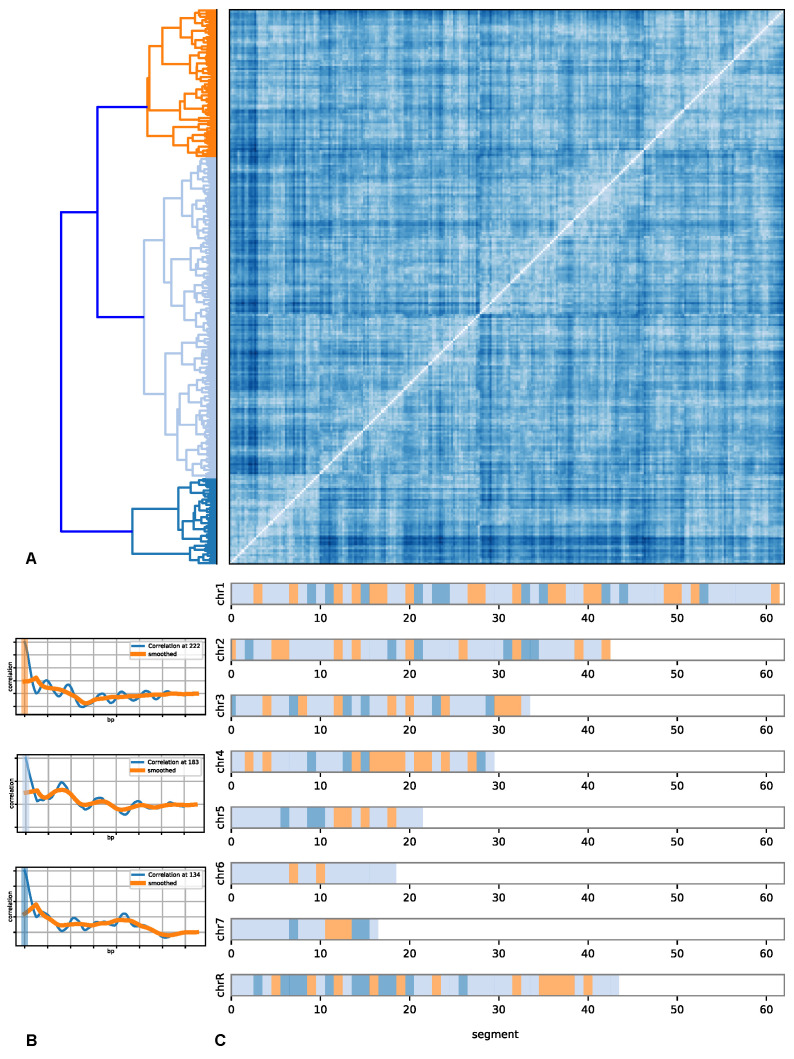
(**A**) shows the genome-wide distance matrix between the correlation functions between segments of size 75 kbp. Hierarchical clustering was applied to identify common patterns. The matrix was sorted according to the patterns. The left side shows the clustering. (**B**) shows the coarse-grained nucleosomal density correlation functions of *Candida albicans* at 5 kb coarse graining. (**C**) shows the genome-wide distribution of segments with colours corresponding to the classification. White space is due to not all chromosomes having the same length. The pattern classification was done genome-wide to yield three main patterns. These three patterns were assigned colours, and the segments of each chromosome corresponding to one of the three patterns are marked. The orange-coloured pattern is characterised by a closely and fairly regularly spaced ordering similar to the tightly packed heterochromatin. The dark and light-coloured blue patterns have lost the regularity and the longer range of the order and thus correspond more to euchromatin. However, note that both these two classes have a huge variety of subclasses. This is not surprising in the sense that one would expect a larger variety of not so ordered patterns in one dimension than for ordered patterns in one dimension.

## Data Availability

Not applicable.

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
