# Peer review of "Superstructure Detection in Nucleosome Distribution Shows Common Pattern within a Chromosome and within the Genome"

_life, 2022, doi:10.3390/life12040541_

Round 1

Reviewer 1 Report

Abstract- Please expand upon the last sentence: what are the implications of understanding the genome beyond hereo- and eu-chromatin and how would that advance the study of chromatin organization?

Page 2: The sentence “Nucleosome positioning is determined by several factors including DNA

sequence, DNA-binding proteins, nucleosome remodelers, RNA polymerases, and more” should be referenced following each of the listed factors.

Page 3: A further discussion on defining the parameter L should be included in the supplementary information with examples of outcomes when L is changed.

Figure 1: Panel A: Include numbering and clearer labeling for the x-axis on each of the graphs. Panel B: Explain the meaning of “liquid-like” in the legend. Panel C: An explanation of the “start” parameter should be added to the legend.

Figure 2: Panel A: A short discussion on the three clusters and their characteristics is necessary in the legend. The dendrogram on the left shows a significant amount of information on the ordering of the classes that are not discussed in detail in the manuscript. Are there observations of note in that data? Panel B: Why was smoothing used and does smoothing result in any loss of information? Panel C: Define each color and the white space.

Page 8: “The orange colored class shows a regular pattern on a short scale whereas the light blue class shows a less stringent regular but still pronounced pattern on a slightly larger scale. The blue colored class shows a rather irregular pattern compared to the other two classes.” Please define these classifications in a less arbitrary manner and include the cutoffs for sorting each section into one of these classes. More explanation is needed for how these classes can be mapped onto hetero- or euchromatin.

Discussion: The conclusions made in this study are rather underwhelming compared to the amount of data analyzed. There is the finding that there are three conserved patterns which can correspond to the hetero- and euchromatin, but there is no further explanation on how these patterns influence genome function and chromatin organization. More analysis on the clustering patterns, comparison with experimental data, and discussion on the applications of the data must be included. For example, SI Figure 5 shows additional clustering patterns that should be discussed on detail. Thus, this manuscript requires major revisions prior to publication.

Author Response

Reviewer’s Comment:

Abstract- Please expand upon the last sentence: what are the implications of understanding the genome beyond hereo- and eu-chromatin and how would that advance the study of chromatin organization?

Our response:

We are grateful for this comment. We added a few more lines in the abstract that explains the understanding of the classification of chromatin into sections on the basis of nucleosome positioning. Moreover, we also added that the pipeline we used is novel in its kind where we transferred a method from condensed matter physics for the first time to help solve the problem of nucleosomal ordering. Here the method is to obtain underlying distribution patterns in nucleosome positioning sequence that governs genomic organization in its locality. The classification obtained here is not trivial, and it lays the foundation of structure detection beyond hetero- and euchromatin.

We added the following text in the abstract.

These non-trivial clusterings correspond to different nucleosome distributions and gene densities governing differential gene expression patterns. Moreover, these distribution patterns inside the chromosome appeared to be conserved throughout the genome and within species. The pipeline of coarse grain nucleosome positioning sequence to identify underlying genomic organisation used in our study is novel and the classifications obtained are unique and consistent.

Reviewer’s Comment:

Page 2: The sentence “Nucleosome positioning is determined by several factors including DNA sequence, DNA-binding proteins, nucleosome remodelers, RNA polymerases, and more” should be referenced following each of the listed factors.

Our response:

We thank the reviewer for pointing out valid referencing of some added factors/text in the manuscript main text. We added the required references 18-25.

Reviewer’s Comment:

Page 3: A further discussion on defining the parameter L should be included in the supplementary information with examples of outcomes when L is changed.

Our response:

Yes, the concept is indeed subtle and needs more explanation. We have added the following text as well as a figure to the Supplementary Information:

Coarse-graining is a procedure that has successfully been developed and applied to condensed matter problems in physics. The basic idea is that each system has a fundamental length scale on which the physical interactions play out. While there are interactions such as excluded volume interaction or Van-der-Waals interactions on a short scale, these all add up to the relevant scale given by the typical correlation length of the system. If the correlations are small, such as in a gas where the constituent particles almost never interact then the fundamental interactions determine the physical scale. For more dense system, there is a scale, the correlation length, on which the system needs to be described.

The coarse-graining procedure is demonstrated in Figure. Panel A shows a noisy signal based on the data shown in panel E. For panels B to D, we increase the coarse-graining length L from 10 to 50 and to 250. The first coarse-graining step shown in panel B already recovers some aspects of the underlying data. The second coarse-graining length L=50 essentially has recovered the underlying structure while for L=250 the signal is too much washed out.

Reviewer’s Comment:

Figure 1:

Panel A: Include numbering and clearer labeling for the x-axis on each of the graphs.

Panel B: Explain the meaning of “liquid-like” in the legend.

Panel C: An explanation of the “start” parameter should be added to the legend.

Our response:

We thank the reviewer for pointing out the discrepancy. We have added the labeling and the following text to the caption.

Panel A shows the performed coarse-graining procedure and results for coarse-graining lengths L of 500 bp, 1000 bp, and 5000 bp.  More structure is visible as L is increased. Going up even further washes out the structure. This is typical for systems with an intrinsic length scale.

Panel B shows the correlation among the coarse-grained super nucleosomes. The structure is that of a system exhibiting short range-order that is liquid-like with first and second nearest neighbor peaks. If there is no order or correlation then the correlation function would be constant. On the other hand, if one would see strong regular peaks, this would indicate a regular ordering with the peak distances giving the preferred distance between the coarse-grained nucleosomes. The oscillatory characteristic with a larger first peak and smaller second peak indicates that two coarse-grained nucleosomes are on average located within a distance from the origin to the first peak and a second coarse-grained nucleosome at the distance indicated by the second peak. Since the peaks are decreasing this ordering diminishes, much like the local ordering in a liquid. On larger scales larger than 50000 bp there is no order, i.e. there is no correlation.

Panel C shows for two chromosomes how the structure differs within as among chromosomes.

The parameter start indicates from where in the chromosomes the structure was computed. One can see that the structure varies within a chromosome, nevertheless, common structures are found.

Reviewer’s Comment:

Figure 2: Panel A: A short discussion on the three clusters and their characteristics is necessary in the legend. The dendrogram on the left shows a significant amount of information on the ordering of the classes that are not discussed in detail in the manuscript. Are there observations of note in that data?  Panel B: Why was smoothing used and does smoothing result in any loss of information? Panel C: Define each color and the white space.

Our response:

We thank the reviewer for pointing out the discrepancy. Yes, indeed, there is a significant more information beyond the three clusters. As there is yet no experimental data available, we have chosen to stop short with the three classes. Nevertheless, the information for the other subclasses is embedded in figures 9 to 16. We have added a movie to the SI that shows some of the subclasses. In this movie, we run through the correlation functions of a single chromosome.

We have added the following text to the caption of figure 2:

Panel C shows the genome-wide distribution of segments with colors corresponding to the classification. White space is due to not all chromosomes having the same length.

The orange colored pattern is characterized by a closely and fairly regularly spaced ordering much like the tightly packed heterochromatin. The dark and light colored blue patterns have lost the regularity and the longer range of the order and correspond thus more to euchromatin. However, note that both these two classes have a huge variety of subclasses. This is not surprising in the sense that one would expect a larger variety of not so ordered pattern in one dimension than for ordered patterns in one dimension.

Reviewer’s Comment:

Page 8: “The orange colored class shows a regular pattern on a short scale whereas the light blue class shows a less stringent regular but still pronounced pattern on a slightly larger scale. The blue colored class shows a rather irregular pattern compared to the other two classes.” Please define these classifications in a less arbitrary manner and include the cutoffs for sorting each section into one of these classes. More explanation is needed for how these classes can be mapped onto hetero- or euchromatin.

Our response:

We rewrite and added text in red in the manuscript to define these classes more sequentially. We also added text in the figure 2 caption to explain it more clearly.

In continuation, the previous comment justifies the sorting and mapping of these obtained classes. “Indeed, there is significantly more information beyond the three clusters. As there is yet no experimental data available, we have chosen to stop short with the three classes. Nevertheless, the information for the other subclasses is embedded in figures 9 to 16. We have added a movie to the Supplementary text that shows some of the subclasses. In this movie, we run through the correlation functions of a single chromosome.”

Main text :

The orange colored class shows a fairly regular pattern and closely spaced ordering on a short scale like tightly packed heterochromatin whereas the light blue class has lost the regularity and shows a less stringent regular but still pronounced pattern on a slightly larger scale. The blue colored class shows a rather very irregular pattern compared to the other two classes and corresponds more to euchromatin.

Figure 2 caption :

Panel C shows the genome-wide distribution of segments with colors corresponding to the classification. White space is due to not all chromosomes having the same length.

The orange colored pattern is characterized by a closely and fairly regularly spaced ordering much like the tightly packed Heterochromatin. The dark and light colored blue patterns have lost the regularity and the longer range of the order and correspond thus more to euchromatin. However, note that both these two classes have a huge variety of subclasses. This is not surprising in the sense that one would expect a larger variety of not so ordered pattern in one dimension than for ordered patterns in one dimension.

Reviewer’s Comment:

Discussion: The conclusions made in this study are rather underwhelming compared to the amount of data analyzed. There is the finding that there are three conserved patterns which can correspond to the hetero- and euchromatin, but there is no further explanation on how these patterns influence genome function and chromatin organization. More analysis on the clustering patterns, comparison with experimental data, and discussion on the applications of the data must be included. For example, SI Figure 5 shows additional clustering patterns that should be discussed on detail. Thus, this manuscript requires major revisions prior to publication.

Our response:

We understand the reviewer’s concern and apologize for not being clear enough. We added text in the abstract to discuss novelty and uniqueness in the classification of chromosomes into similar structurally organized patterns. We also stated that there is yet no experimental data available, we have chosen to stop short with the three classes. the information for the other subclasses is embedded in figures 5 to 16. But, these patterns obtained are generally coincident with the experimentally found gene densities that lead to genome spatial organization and genome function.

We have rewritten and added the following text in the discussion to emphasize the results of clustering patterns and their application.

Around five patterns of chromosomal organization are obtained for each chromosome by analyzing the nucleosome positioning data distribution. These patterns obtained are generally coincident with gene densities and lead to the distinct spatial organization of genomic DNA. The genome’s hierarchical structure-function relationship (12) is governed by chromatin domains and their higher-order folding. The formation of chromatin boundaries and associated TADs are controlled by the nucleosome distribution patterns. Recent studies by Wiese et. al. (16) suggested that domain formation and genome organization can be predicted with nucleosome positioning only. Pulivarty et. al. (44) primarily focused that nucleosome studies are limited to a very local individual promoter and enhancer but can be a more general mechanism by which cells can regulate the accessibility of the genome during development at different scales. After an extensive analysis of nucleosome positioning data, the way of organization of nucleosomal distribution patterns is found to be different at different scales and for different chromosomes. The distinct patterns obtained from our calculation correspond to different ways of nucleosome positioning and may control domain formation and genome organization in the cell. However, the three distinct patterns of nucleosome organization that appeared to be conserved in the genome show the global consistency of distribution patterns inside the genome. The consistency in different kinds of distinct patterns observed in the genome corresponds to identical gene densities and similar expression regions for specific locations inside the cell.

Reviewer 2 Report

This article provides bioinformatic analyses from previously reported data on nucleosomal positioning in Candida albicans. Using a number of computational approaches, several patterns of chromosomal organization are defined based on the data, generally coincident with distinct gene densities. These three patterns are consistent between chromosomes. This is an interesting observation; however, I am concerned about the novelty of the study, which I find the main flaw of this manuscript.

Also, it would be interesting to know whether in other species these patterns can be identified using the same pipeline.

Finally, I suggest improving the discussion section by including additional references to the studies on genome architecture and how these fit with the results.

Author Response

Reviewer’s Comment:

This article provides bioinformatic analyses from previously reported data on nucleosomal positioning in Candida albicans. Using a number of computational approaches, several patterns of chromosomal organization are defined based on the data, generally coincident with distinct gene densities. These three patterns are consistent between chromosomes. This is an interesting observation; however, I am concerned about the novelty of the study, which I find the main flaw of this manuscript.

Our response:

We thank the reviewer for the positive comment. Yes, the obtained patterns are not trivial and are very unique, and are consistent between chromosomes. Moreover, this consistency prevails among species too. We emphasized that the pipeline we used is novel in its kind where we used a method of condensed matter physics to solve problems in the biological system. Here the method is to obtain underlying distribution patterns in nucleosome positioning sequence that governs genomic organization in its locality. The classification obtained here is not trivial, and it lay the foundation of superstructure detection beyond hetero- and euchromatin.

We added a few more lines in the abstract and discussion to explain this better.

Abstract: These non-trivial clusterings correspond to different nucleosome distributions and gene densities governing differential gene expression patterns. Moreover, these distribution patterns inside the chromosome appeared to be conserved throughout the genome and within species. The pipeline of coarse grain nucleosome positioning sequence to identify underlying genomic organization used in our study is novel and the classifications obtained are unique.

Discussion:

Around five patterns of chromosomal organization are obtained for each chromosome by analyzing nucleosome positioning data distribution. These patterns obtained are generally coincident with gene densities and lead to the distinct spatial organization of genomic DNA. Genome’s hierarchical structure-function relationship (12) is governed by chromatin domains and their higher-order folding. The formation of chromatin boundaries and associated TADs are controlled by the nucleosome distribution patterns. Recent studies by Wiese et. al. (16) suggested that domain formation and genome organization can be predicted with nucleosome positioning only. Pulivarty et. al. (44) primarily focused that nucleosome studies are limited to a very local individual promoter and enhancer but can be a more general mechanism by which cells can regulate the accessibility of the genome during development at different scales. After an extensive analysis of nucleosome positioning data, the way of organization of nucleosomal distribution patterns is found to be different at different scales and for different chromosomes. The distinct patterns obtained from our calculation correspond to different ways of nucleosome positioning and may control domain formation and genome organization in the cell. However, the three distinct patterns of nucleosome organization that appeared to be conserved in the genome show the global consistency of distribution patterns inside the genome. The consistency in different kinds of distinct patterns observed in the genome corresponds to identical gene densities and similar expression regions for specific locations inside the cell.

Reviewer’s Comment:

Also, it would be interesting to know whether in other species these patterns can be identified using the same pipeline.

Our response:

We give special thanks to the reviewer for this comment. Yes, indeed this pipeline is valid for any species subjected that their whole genome has been sequenced. However, we limited our study to the species Candida albicans due to its comparatively smaller genome, and also its whole genome has been sequenced.

We added the following text in the manuscript to explain this clearly.

The partitioning into a clear pattern, genome-wide is not limited to species Candida albicans but the pipeline is generalized and can be used for any species in which the whole genome has been sequenced.

Reviewer’s Comment:

Finally, I suggest improving the discussion section by including additional references to the studies on genome architecture and how these fit with the results.

Our response:

Yes, an improvement and explanation were needed. We have rewritten and added the following text in the discussion to explain it much better. I also added some related references for better understanding.

Around five patterns of chromosomal organization are obtained for each chromosome by analyzing nucleosome positioning data distribution. These patterns obtained are generally coincident with gene densities and lead to the distinct spatial organization of genomic DNA. Genome’s hierarchical structure-function relationship (12) is governed by chromatin domains and their higher-order folding. The formation of chromatin boundaries and associated TADs are controlled by the nucleosome distribution patterns. Recent studies by Wiese et. al. (16) suggested that domain formation and genome organization can be predicted with nucleosome positioning only. Pulivarty et. al. (44) primarily focused that nucleosome studies are limited to a very local individual promoter and enhancer but can be a more general mechanism by which cells can regulate the accessibility of the genome during development at different scales. After an extensive analysis of nucleosome positioning data, the way of organization of nucleosomal distribution patterns is found to be different at different scales and for different chromosomes. The distinct patterns obtained from our calculation correspond to different ways of nucleosome positioning and may control domain formation and genome organization in the cell. However, the three distinct patterns of nucleosome organization that appeared to be conserved in the genome show the global consistency of distribution patterns inside the genome. The consistency in different kinds of distinct patterns observed in the genome corresponds to identical gene densities and similar expression regions for specific locations inside the cell.

Reviewer 3 Report

The study "Superstructure detection in nucleosome distribution shows common pattern within a chromosome and within the genome" provides an interesting aspect to the field an takes on a holoistic approach. The study is sound and conducted in an excellent manner. However, important points in background is lacking. Positioned nucleosomes are biologically relevant and seem to be established upon transcriptional state of a region. This is clearly seen in development where certain region suddenly forms positioned nucleosome, and the trigger is not fully understood. This dynamic nature of positioned nucleosomes are not discussed in the introduction or in the discussion. It is also important to choose the right model for the study, and be aware of the limitations in the choice. The model, C. albecans, is in essense a unicellular organism and may have specific chromatin activities. Do this fungus have DNA methylation or histone H1 operating in the same way as mammals or insects? This is not discuss, nor is the implication of having found several differnt states. Several chromatin states have previously been  reported and how do the ones discovered here relate to these previously found.

The MNase assay also have problems with titration - how have the authors addressed this.

Author Response

We thank the reviewer for examining our manuscript thoroughly and adding constructive comments. We have referred to the reviewer’s comments and made significant modifications in the manuscript accordingly and where needed.

Response to the reviewer’s comments on the manuscript:

=============================================================

Reviewer’s Comment:

The study "Superstructure detection in nucleosome distribution shows common pattern within a chromosome and within the genome" provides an interesting aspect to the field an takes on a holoistic approach. The study is sound and conducted in an excellent manner. However, important points in background is lacking.

Positioned nucleosomes are biologically relevant and seem to be established upon transcriptional state of a region. This is clearly seen in development where certain region suddenly forms positioned nucleosome, and the trigger is not fully understood. This dynamic nature of positioned nucleosomes are not discussed in the introduction or in the discussion.

Our response:

Thank you for pointing out the possible scopes of the current manuscript. Indeed, Nucleosome dynamics are well studied, where it performs breathing, sliding, gapping, and repositioning to give transient access to DNA Binding Proteins (DBPs) to perform cellular metabolic functions like transcription, translation, gene regulation and so on. However, due to experimental limitation the nucleosome positioning is a sequence based mapping approach that provides a static picture of nucleosome position in a cell- and time-averaged manner. The nucleosome positioning snapshot at different times (if available in future) can be used to identify patterns that may explain the trigger mechanism of nucleosome positioning. Few studies have reported the regular positioning of nucleosomes triggered at transcription start sites (TSSs) and CTCFs binding sites, but our study is more keen towards finding the underlying mechanical code that govern the nucleosome positioning distribution pattern along the chromosomes/genome. Moreover, these dynamics of nucleosomes are limited to a few bps, but the superstructure we found here is at the scale of ~75kbps which rules out the possible small scale dynamics to capture major changes that occur at different stages of the cell cycle.

In the iNPS database which is used here to get nucleosome positioning data, nucleosome positions are averaged over cell- and time, where they implicitly consider the dynamics of nucleosomes in the form of snapshots taken at different intervals. Afterwards, the peaks are annotated to get possible nucleosomes position. The higher the peaks, the  higher the probability of occurrence of nucleosomes.

However, a little more explanation was needed so we add the following in the introduction of the main text along with the previous explanation.

Although nucleosome positioning is a dynamic process, the sequence-based mapping approach identifies its position only in a cell- and time-averaged manner. The technology of micrococcal nuclease (MNase) digestion combined with high-throughput sequencing (MNase-seq) is a powerful method to map the genome-wide distribution of nucleosome positioning and its occupancy. The resulting occupancy maps are ensemble averages of heterogeneous cell populations. However, it is necessary to retrieve the cell specific features from the population average to reveal the mechanism of nucleosome organisation and its translocation along the genome. Zhang et al. has developed an algorithm called "Nucleosome Positioning from Sequencing" (NPS) to predict accurate nucleosome positioning from the MNase-seq data, which was later improved to iNPS (improved NPS). The nucleosome positioning is considered as an averaged static picture where they implicitly consider the nucleosome dynamics in the form of snapshots at different time- and cell- average. This nucleosome positioning provides the frequency of its occurrence from which peaks are annotated to obtain possible nucleosome location along the sequence. In short, the nucleosome positioning data from iNPS are the most probable nucleosome position along the chromosome.

Reviewer’s Comment:

It is also important to choose the right model for the study, and be aware of the limitations in the choice. The model, C. albecans, is in essense a unicellular organism and may have specific chromatin activities. Do this fungus have DNA methylation or histone H1 operating in the same way as mammals or insects?

Our response

The reviewer is right in his assessment that the choice matters. What we had in mind selecting Candida albicans was to take a very simple completely sequenced organism that is small enough to be computationally viable. Furthermore, the organism should allow for similar mechanisms that are found in eucaryotes. Whereas these are technical decisions, we also wanted to select a species that should have a clinical prevalence.

We added the following to the introduction:

We have chosen candida albicans as a simple completely sequenced organism that is small enough to be computationally viable. Furthermore, C. albicans allows for similar mechanisms that are found in eucaryotes. Indeed epigenetic mechanisms across animals, plants, and fungi include DNA methylation as a common epigenetic signaling mechanism and is present in C. albicans. A putative histone H1 has been identified \cite{Skrzypek:2017wq}. Whereas these are technical decisions, we also wanted to select a species that should have a clinical prevalence.

And the following reference

\bibitem{ Skrzypek:2017wq} Skrzypek, Marek S. and Binkley, Jonathan and Binkley, Gail and Miyasato, Stuart R. and Simison, Matt and Sherlock, Gavin, The Candida Genome Database (CGD): incorporation of Assembly 22, systematic identifiers and visualization of high throughput sequencing data, Nucleic Acids Research, 2017, 45, D592--D596

Reviewer’s Comment:

This is not discuss, nor is the implication of having found several differnt states. Several chromatin states have previously been  reported and how do the ones discovered here relate to these previously found.

Our response

We are aware that there are many possible chromatin states beyond heterochromatin and euchromatin as solenoid \cite{Finch-1976}, zig-zag ribbon \cite{Diesinger-2010} or other irregular structures reported in the literature \cite{Williams-1986}. However, these precise chromatin states depend of parameters as pH and temperature as well as the presence of CTCF or other molecules. These cross-correlation study is out of the scope of the present manuscript, but they will be addressed in future works.

We added the following

These sub-features compose elaborated chromatin states as solenoid \cite{Finch-1976}, zig-zag ribbon \cite{Diesinger-2010} or other structures \cite{Williams-1986}, which demand a cross-correlation analysis with CTCF binding sites \cite{Oiwa-2022}, CpG island position \cite{Gardiner-1987} and other data.

and the references.

\bibitem{Finch-1976} Finch JT, Klug A. Solenoidal model for superstructure in chromatin. Proc. Natl. Acad. Sci. USA 1976; 73: 1897-1901.

\bibitem{Diesinger-2010} Diesinger PM, Kunkel S, Langowski, Heermann DW. Histone Depletion Facilitates Chromatin Loops on the Kilobasepair Scale. Biophysical Journal 2010; 99:2995-3001.

\bibitem{Williams-1986} Williams SP, Athey BD, Muglia LJ, Schappe RS, Gough AH, Langmore JP. Chromatin Fibers are Left-Handed Double Helices with Diameter and Mass per Unit Length that Depend on Linker Length. Biophys. J. 1986; 49: 233-248.

\bibitem{Oiwa-2022} Oiwa NN, Li K, Cordeiro CE, Heermann DW. Physical Biology 2022, accepted for publication.

\bibitem{Gardiner-1987} Gardiner-Garden M, Frommer M. CpG Islands in Vertebrate Genome. J. Mod. Biol. 1987; 196: 261-282.

Reviewer’s Comment:

The MNase assay also have problems with titration - how have the authors addressed this.

Our response

Yes, excellent point raised by the reviewer. The reason we have not dealt with the titration issue is that we are dependent on publicly available data. The data that we have access to does not deal with the issue.

We added the following to the introduction:

… The resulting occupancy maps are ensemble averages of heterogeneous cell populations and may also be influenced by titration \cite{Mieczkowski:2016uo}.

and the references.

\cite{Mieczkowski:2016uo} Mieczkowski, Jakub and Cook, April and Bowman, Sarah K. and Mueller, Britta and Alver, Burak H. and Kundu, Sharmistha and Deaton, Aimee M. and Urban, Jennifer A. and Larschan, Erica and Park, Peter J. and Kingston, Robert E. and Tolstorukov, Michael Y., MNase titration reveals differences between nucleosome occupancy and chromatin accessibility, Nature Communications, 2016, 7, 11485

Round 2

Reviewer 1 Report

The manuscript revision now has noticeable improvements over the original. It is clear now what the possibilities and limitations of this computational study has and does not have. With a few minor revisions, this manuscript should be published in Life.   

Page 3- Please explain the clinical relevance of C. albicans or explain how these data/methods can be extrapolated to larger genomes that are more clinically relevant. It is concerning that one of the reasons why this species was chosen was for computational viability so a discussion on this subject may be necessary for future reference.

  SI Figure 28 caption is missing.  Also, please make sure spelling and grammar are correct- for example, "eucaryotes" on page 3.